# Mechanical Behaviors of Microalloyed TRIP-Assisted Annealed Martensitic Steels under Hydrogen Charging

**DOI:** 10.3390/ma14247752

**Published:** 2021-12-15

**Authors:** Xiongfei Yang, Hao Yu, Chenghao Song, Lili Li

**Affiliations:** 1School of Materials Science and Engineering, University of Science and Technology Beijing, Beijing 100083, China; yxf@pgvrd.com (X.Y.); songch@dgut.edu.cn (C.S.); lilili0115@126.com (L.L.); 2School of Mechanical Engineering, Dongguan University of Technology, Dongguan 523808, China

**Keywords:** mechanical property, microalloying, TAM steel, hydrogen charging, thermal desorption, SSRT

## Abstract

Transformation Induced Plasticity (TRIP)-assisted annealed martensitic (TAM) steel sheets with various microalloying additions such as niobium, vanadium, or titanium were prepared on laboratory scale and subjected to a double-quenching and austempering heat treatment cycle. Slow strain rate tensile (SSRT) was tested on the investigated TAM steels with and without hydrogen charging to reveal their tensile behaviors and hydrogen induced embrittlement effects. Microstructure observations by scanning electron microscope (SEM) are composed of a principal annealed martensitic matrix and 11.0–13.0% volume fraction of retained austenite, depending on the type of microalloying addition in the different steels. SSRT results show that these TRIP-assisted annealed martensitic steels under air media conditions combine high tensile strength (>1000 MPa) and good ductility (~25%), while under hydrogen charging condition, both tensile strength and ductility decrease where tensile strength ranges between 680 and 760 MPa, down from 1000–1100 MPa, and ductility loss ratio is between 78.8% and 91.1%, along with a total elongation of less than 5%. Hydrogen charged into steel matrix leads to the appearance of cleavage fractures, implying the occurrence of hydrogen induced embrittlement effect in TAM steels. Thermal hydrogen desorption results show that there are double-peak hydrogen desorption temperature ranges for these microalloyed steels, where the first peak corresponds to a high-density dislocation trapping effect, and the second peak corresponds to a hydrogen trapping effect exerted by microalloying precipitates. Thermal desorption analysis (TDS) in combination with SSRT results demonstrate that microalloying precipitates act as irreversible traps to fix hydrogen and, thus, retard diffusive hydrogen motion towards defects, such as grain boundaries and dislocations in microstructure matrix, and eventually reduce the hydrogen induced embrittlement tendency.

## 1. Introduction

Advanced high strength steels (AHSSs), due to their high performance and capability of satisfying with increasing environmental-friendly requirements, have seen extensive application in various fields such as engineering, machinery, and automobiles [1,2]. With the increasing use of AHHSs, delayed fracture induced by hydrogen invasion becomes a challenging issue, since AHHSs are prone to hydrogen invasion and diffusion towards the metal matrix when they have been subjected to cold-worked process or serviced in moist atmosphere. Extensive efforts have been made to eliminate this phenomenon of hydrogen induced delayed fracture behavior in AHHSs [3]. Previous works [4,5,6,7,8,9,10] showed that after the introduction of various kinds of hydrogen traps into a steel matrix, the hydrogen induced delayed fracture behavior in high strength steels could be alleviated. Effective hydrogen traps, for example, structure defects like grain boundary and dislocation, retained austenite, MnS inclusion, and microalloying precipitates such as NbC, TiC, and NbN, have been intensively investigated. They are classified into reversible or irreversible traps depending on their ability to absorb hydrogen, as indexed by activation energy [4,5,6,7].

Transformation Induced Plasticity (TRIP)-assisted annealed martensitic (TAM) steels were initially developed by Sugimoto et al. [11]. Although TAM steels show an excellent combination of high strength and good ductility, they are highly prone to hydrogen induced delayed fracture due to low hydrogen atom solubility in their martensitic matrix [12]. Until now, reported publications hardly focused on hydrogen embrittlement of TAM steel, and much less on how to improve its hydrogen induced delayed fracture behavior [4,10,11]. Thus, detailed understanding hydrogen trapping behavior in TAM steels and their hydrogen induced embrittlement performance is essential to commercialize the steel for certain applications, such as automobile structural parts.

In TAM steels, retained austenite plays an effective role in improving ductility due to the so-called TRIP effect [13]. TAM steels would be microalloyed by elements such as niobium, titanium, and vanadium to further improve their certain properties. Among these microalloying elements, only V can precipitate at a tempering temperature of ~400 °C. The low-temperature dispersedly precipitated VC particles, not only further enhance strength, but improve resistance to hydrogen induced embrittlement [11] of TAM steels. Recent results [10,14,15,16,17] on vanadium-added steels suggested that low temperature vanadium precipitates, usually in form of V_4_C_3_, could effectively produce a higher resistance to hydrogen induced delayed fracture for ultra-high strength steels. Meanwhile, TiC particles have been well established as strong hydrogen trapping sites. Furthermore, hydrogen atoms have been directly observed in the non-coherent interface between NbC precipitates and its adjacent matrix [18].

In this work, hydrogen trapping behaviors and their effect on the mechanical properties of four TAM steels with various microalloying additions, such as Nb, Ti, and V, had been investigated through electrochemical hydrogen charging process, slow strain rate tensile (SSRT) tests, microstructure observation, and thermal hydrogen desorption analysis (TDS). The reasons for selecting TAM steels in this study are as follows: its martensitic matrix is highly prone to hydrogen embrittlement, and presence of a certain volume fraction of austenite in the martensitic matrix enables it to capture diffusive hydrogen atom.

## 2. Materials and Methods

### 2.1. Materials Preparation

The reference material with a composition of 0.2% C–1.50% Si–2.0% Mn (wt.%), and three other materials microalloyed by Nb, Ti, or V addition into reference steel are comparatively studied in this work. Similar content of an individual microalloying element is added in the three corresponding steels. All four heats were prepared in a laboratory-scale vacuum induction furnace and casted into 50 kg ingots for individual heat. The ingots were reheated up to 1200 °C, isothermal holding for 1 h, and subsequently hot-rolled into 4 mm thick plates through several passes with finished rolling temperature at 800 °C. Then, the hot rolled plates were subjected to pickling treatment to remove oxide scales on the plate surface, and finally cold rolled into sheets of 2 mm thickness. The detailed chemical compositions of the studied steels are given in Table 1.

To obtain the required microstructure consisting of a principal annealed martensite matrix in combination with a certain amount of retained austenite, the cold-rolled sheets were subjected to specially designed heat-treatment cycles, as indicated in Figure 1. All applied heating rates and cooling (quenching) rates in this work were at 10 °C/s and 50 °C/s, respectively.

### 2.2. Slow Strain Rate Tensile (SSRT) Tests

SSRT had been evaluated at room temperature on a Xi’an LETRY stress-corrosion cracking tester manufactured by a Chinese provider. Tensile specimens with dimensions of 140 mm gauge in length and 30 mm gauge in width were machined from the TAM steel sheets paralleling to rolling direction. Before conducting the SSRT test, a 500 N pre-tensile load was applied to both ends of the specimen to eliminate any potential gap occurring between an under-tested specimen and the tester clamp. The applied constant strain rate was kept at 1 × 10^−6^/s. For different steels, single SSRT test was carried out for each condition, that is, under air media or hydrogen charging state. For the hydrogen charging condition, the testing specimen was placed into 0.5 mol/L H_2_SO_4_ electrolyte mixed with 0.25 g/L NaAsO_3_ solution [20]. The remaining section, except for the testing gauge, was covered by a type of 704 Silica gel to protect it from hydrogen invasion. During the SSRT test, the specimen was continuously subjected to an electro-chemical hydrogen charging process. Additional experiments conducted by the authors showed that under a lower applied charging current, the amount of hydrogen charged into the specimen was at a low value. On the other hand, a much higher charging current would lead to crack initiation on the specimen surface. In this work, the applied cathodic charging current was chosen at 5.0 mA/cm^2^. Therefore, during the whole SSRT test, the charging current was maintained at 5.0 mA/cm^2^ until specimen fractured.

### 2.3. Hydrogen Analysis

Before the thermal hydrogen desorption analysis, the samples were electrical-chemically charged to perform a hydrogen charging test in standard Devanathan–Stachurski double electrode cells, shown in Figure 2. The tested specimen was a disc, 0.3 mm in thickness. Again, single test was carried out per specimen, regardless of whether it was subjected to the hydrogen charging process or not.

The anode side of the specimen was chemically deposited by a layer of 200 nm nickel film to protect the anode surface from dissolution. Electro-chemically charging hydrogen was carried out at room temperature in 0.5 mol/L H_2_SO_4_ electrolyte with the addition of 0.22 g/L thiourea (CH_4_N_2_S) solution [21]. The charging hydrogen process was lasted for 12 h at an applied cathodic charging current of 10.0 mA/cm^2^, instead of 5.0 mA/cm^2^ used in the SSRT test to maximize the finally charged hydrogen content into the sample. Once the charging process had finished, the corrosion product was mechanically removed from the surface of the hydrogen charged disc. Then it was immediately cleaned with de-ionized water for the purpose of further thermal desorption analysis (TDA).

TDA experiments were conducted on the hydrogen charged disc by a UK manufactured Markes TD100 thermal desorption spectrometry. Sampling time interval was selected at 5-mins and the heating rate was kept at 100 °C/h under vacuum conditions during the thermal desorption test. The investigated temperatures in this work ranged from 40 to 800 °C. A triple quadrupole series mass spectrometry was used to detect the hydrogen desorption rate. The desorption hydrogen weight (content) was calculated from the measured desorption curves through a line integral method.

### 2.4. Analysis of Microstructure and Fracture

The microstructure of TAM steels and fracture appearance of the samples after the SSRT test were observed by a JEM 2100F Scanning Electron Microscope (SEM). Volume fractions of the retained austenite for various kinds of TAM steels were measured by a DMAX-RB X-rays diffraction. The applied working parameters for XRD measurement are as follows: λ = 0.15406 nm, working voltage at 40 kV and current at 150 mA, step size of 0.02°, scanning speed of 1°/min, and scanning range (2θ) of 37–93°.

## 3. Results

### 3.1. Microstructure Observations

Microstructure observations for four kinds of steels after TAM treatments are shown in Figure 3. From Figure 3, the regular annealed martensite laths are visible. Observed microstructures are composed of a principal annealed martensitic matrix and 11.0–13.0% volume fraction of retained austenite measured by XRD technique, depending on the type of microalloying additions in different kinds of steels. The retained austenite is distributed between annealed martensite laths, as shown in Figure 3e.

Although the steels TAM-V and TAM-R show a slightly higher volume fraction of retained austenite compared with those of steels TAM-Ti and TAM-Nb, no significant difference in microstructure has been observed in the studied steels.

### 3.2. Stress-Strain Curves during SSRT Test

The obtained engineering stress–strain curves of four studied steels during the SSRT tests under hydrogen charging and no charging conditions are shown in Figure 4. It is clearly shown in Figure 4 that when samples are subjected to electro-chemically hydrogen charging during tensile test, the tensile strength decreases and significant reduction in total elongation is seen for all steels.

The SSRT test results for four studied steels under both conditions of air media (no hydrogen charging) and electrolyte (hydrogen charging) are shown in Table 2. Based on the measured SSRT tensile data, hydrogen induced tensile strength loss ratio *I_T_* can be calculated, referring to Equation (1), and calculated ductility loss ratio *I_D_*, referring to Equation (2) are generated.
*I_T_* = (*R_m_^o^* − *R_m_*)/*R_m_^o^* × 100%(1)
where *R_m_^o^* is the tensile strength for the specimen during an SSRT test under air media, and *R_m_* is the SSRT tensile strength for the hydrogen charged specimen of the same steel.
*I_D_* = (*A^o^* − *A*)/*A^o^* × 100%(2)
where *A^o^* is the total elongation for the specimen during an SSRT test under air media, and *A* is the SSRT total elongation for the hydrogen charged specimen of the same steel.

Although all steels show declining tensile properties when the samples are subjected to electro-chemically hydrogen charging during the SSRT test, the loss values are different for different steels. TAM-R steel shows the biggest ductility loss, where its relative reduction value *I_D_* is 91.1%. It gives the lowest tensile strength loss ratio *I_T_*. All three kinds of microalloyed TAM steels show smaller ductility loss ratios than that of microalloying element free steel TAM-R. For three different TAM steels exposed to the hydrogen charging process, significant difference in their ductility loss and low elongation value can be noted. Obviously, TAM-Ti steel shows the highest elongation value and the smallest ductility loss ratio among the three kinds of TAM steels exposed to the hydrogen charging process.

### 3.3. Fractural Appearance

The SSRT for all four kinds of steels was evaluated under hydrogen charging conditions or air media. The fracture appearance of each specimen was observed using a JEM 2100F SEM and shown in Figure 5. Figure 5a,c,e,g indicate that a considerable number of dimples are present in the fracture appearance of no hydrogen charged steels after SSRT tests, showing a typical ductile fracture pattern, while in Figure 5b,d,f,h, the dimples disappear and cleavage faces are present in the fracture appearance of hydrogen charged steels after the SSRT test. Obviously, diffusive hydrogen charged into steel matrix leads to the fracture pattern changing from the ductile fracture for steels without hydrogen charging, into the cleavage fracture when the steel are exposed to hydrogen charging during tensile tests.

### 3.4. TDA Results

The thermal hydrogen desorption rate vs heating temperature during the process of heating up to 800 °C for the four hydrogen-charged steels is plotted in Figure 6, where the incorporated chart is the enlarged second-peak section shown in the corresponding chart. Except for steel TAM-R, the microalloyed steels show a double-peak curve, where the temperature for the so-called bigger first-peak section ranges between 160 °C and 190 °C, and the temperature for the smaller second-peak section ranges between 245 °C and 305 °C. For steel TAM-V, the first and second peak temperatures are 167 °C and 305 °C, respectively. For steels TAM-Ti, the first and second peak temperatures are 179 °C and 300 °C, respectively. For steel TAM-Nb, the first and the second peak temperatures are 168 °C and 245 °C, respectively. There is a smaller second peak in the case of TAM-Nb steel compared with those of TAM-V and TAM-Ti steels. For reference steel TAM-R, only the single peak temperature at 183 °C can be observed.

From the TDA curves, the desorption hydrogen content at individual section can be separated from each other. The hydrogen contents during thermal desorption for different hydrogen-charged steels are shown in Table 3. During the first peak section, total desorption hydrogen contents for TAM-V, TAM-Ti, TAM-Nb and TAM-R steels are 8.66 ppm, 9.43 ppm, 9.34 ppm and 9.02 ppm, respectively. The desorption hydrogen contents during the second peak section for the above-mentioned steels are 0.42 ppm, 0.22 ppm, 0.10 ppm and 0.02 ppm, respectively.

## 4. Discussion

We can see from Table 3 that the reference steel TAM-R has the highest ductility value but the lowest tensile strength value among the four studied steels without electro-chemically hydrogen charging. The reason behind this is the TRIP effect of retained austenite. The TRIP effect contributes to good ductility during the tensile test. On the other hand, it is well known that precipitates in the matrix lead to a precipitation strengthening effect, but they deteriorate the ductility and toughness of the steels. Although the TAM-V steel presents a similar volume fraction of retained austenite compared to the reference steel, there is no doubt that large amounts of vanadium precipitates exist in the former steel, which will be discussed in detail in the following context. Thus, it is reasonably concluded that the retained austenite in combination with precipitates determines the tensile behaviors of the studied steels under no hydrogen charging conditions.

For all kinds of steels, both tensile strength and total elongation decline when samples are exposed to electro-chemical hydrogen charging during SSRT test. Particularly for the reference steel without a microalloying addition, it gives the highest volume fraction retained austenite and is theoretically free of microalloying precipitates, but produces a high hydrogen induced ductility loss ratio, when compared with the other three microalloyed steels.

During the SSRT test, diffusive hydrogen will readily segregate towards stress concentration tips existing in the structure matrix and result in reducing the resistance to HIC formation. Furthermore, the segregated hydrogen atoms will accumulate into gaseous hydrogen, and then exert additional stress on the stress concentration tip, which leads to appearance of micro-cracks and cleavage fracture [22]. It is expected that hydrogen motion in the matrix dominates the tensile behaviors of hydrogen charged steels, that is, causing significant ductility reduction.

Varieties of defects in steel matrix such as grain boundaries, dislocations, and precipitates can play important roles in hydrogen diffusion behavior in steels. The four studied steels have a similar base chemistry and are subjected to the same heat treatment cycle histories. As discussed in earlier work [19], hydrogen atoms cannot get into the interface between the retained austenite and matrix; thus, the amount of hydrogen absorbed by retained austenite is assumed to be at a low value. This means that the amount of hydrogen in transformed martensite after TRIP effect during tensile process is limited. Although retained austenite in high strength steel can act as an effective trap site [23], in this work, retained austenite distributed in the studied steels matrix is expected to have no negative effect on hydrogen induced embrittlement.

For all four steels, TDS results show a bigger first-peak temperature ranging from 160 °C to 190 °C. Since all steels were subjected to a certain heat treatment cycle to obtain the designed TAM-like microstructure, it is reasonable to assume that high density dislocation exists in the martensite structure matrix. Thermal desorption hydrogen in this peak temperature range corresponds to hydrogen absorbed/trapped by high density dislocations distributed in martensitic structures [18].

For the microalloyed steels, there is smaller second-peak temperature ranging from 245 °C to 305 °C, where thermal desorption hydrogen amount was 10–100 times smaller than those of the bigger first -peak section. Nonetheless, these values are far greater than that of TAM-R steel with regard to their orders. Chen [18] utilized Cryo-transfer atom probe tomography to directly observe hydrogen existence in the non-coherent interface between NbC precipitates and their adjacent matrix, which provided the evidence that non-coherent interfaces between a microalloying precipitate and matrix could act as effective traps to fix hydrogen.

In this study, Nb, V, and Ti elements could precipitate in steels during heat treatment cycles. These precipitates probably act as irreversible trapping sites to absorb small amount of diffusive hydrogen, and then retard the diffusive hydrogen motion towards grain boundaries and especially dislocation, that is, lowering the possibility of generating stress concentration on the so-called defect sites, eventually reducing the hydrogen embrittlement tendency.

It seems that when the amount of hydrogen trapped by microalloying precipitates is higher, the hydrogen induced ductility loss ratio, is lower. However, when comparing the desorption hydrogen contents during the second-peak section for TAM-V and TAM-Ti steels, there is a different story. Although desorption hydrogen content from precipitates in TAM-V steel is higher than that of TAM-Ti steel, the former is slightly more prone to hydrogen embrittlement than the latter in terms of hydrogen induced ductility loss ratio. The reason for this is not yet clear and needs to be clarified in further study.

## 5. Conclusions

TAM steels with various kinds of microalloying additions were prepared by a specially designed double-quenching and austempering heat treatment cycle. The SSRT were evaluated on the obtained TAM steels under hydrogen charging and no hydrogen charging conditions to reveal their tensile behaviors and hydrogen induced embrittlement effects. The important conclusions obtained are as follows:TAM steels under no hydrogen charging conditions show an excellent combination of high tensile strength (>1000 MPa) with good ductility (~25%). Under hydrogen charging condition, both tensile strength and ductility decline, where tensile strength decreases from 1000–1100 MPa under no hydrogen charging condition to 680–760 MPa, and the observed significant ductility loss ratios are between 78.8% and 91.1%, with total elongation even less than 5%.The tensile behaviors of TAM steels under no hydrogen charging conditions are determined by the volume fraction of retained austenite and microalloying precipitates, showing a typical ductile fracture pattern. Hydrogen charged into the steel matrix leads to a cleavage fracture pattern, implying the occurrence of a hydrogen induced embrittlement effect.Thermal hydrogen desorption results show that there are double-peak hydrogen desorption temperature ranges for the studied microalloyed steels, where the first peak is assumed to relate to the high-density dislocation trapping effect, and the second peak corresponds to the hydrogen trapping effect exerted by microalloying element precipitates. There is no apparent second peak in the case of reference TAM steel since no microalloying precipitates are expected to form in the microstructure.Sites such as microalloying precipitates in steel are assumed to act as effective traps to fix hydrogen atoms and, thus, retard hydrogen motion towards grain boundaries and especially dislocations, lowering the possibility of causing stress concentration among the so-called defect sites, and eventually reducing hydrogen induced embrittlement tendency in TAM steels.

## Figures and Tables

**Figure 1 materials-14-07752-f001:**
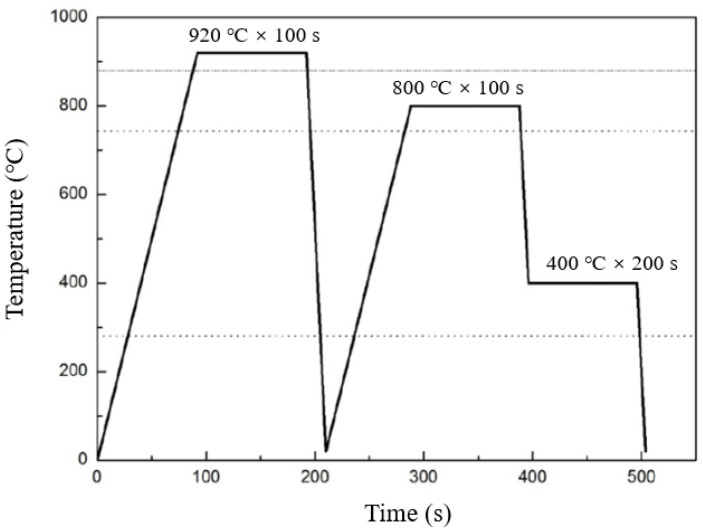
Heat treatment cycles applied for producing TAM steels [19].

**Figure 2 materials-14-07752-f002:**
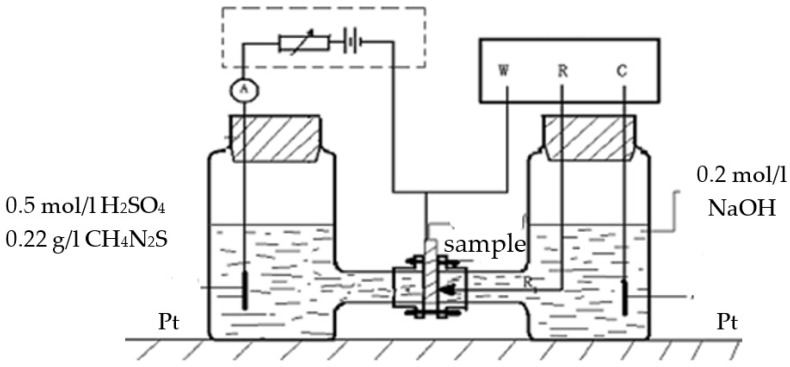
Devanathan–Stachurski double electrode cells used for the hydrogen charging process [19].

**Figure 3 materials-14-07752-f003:**
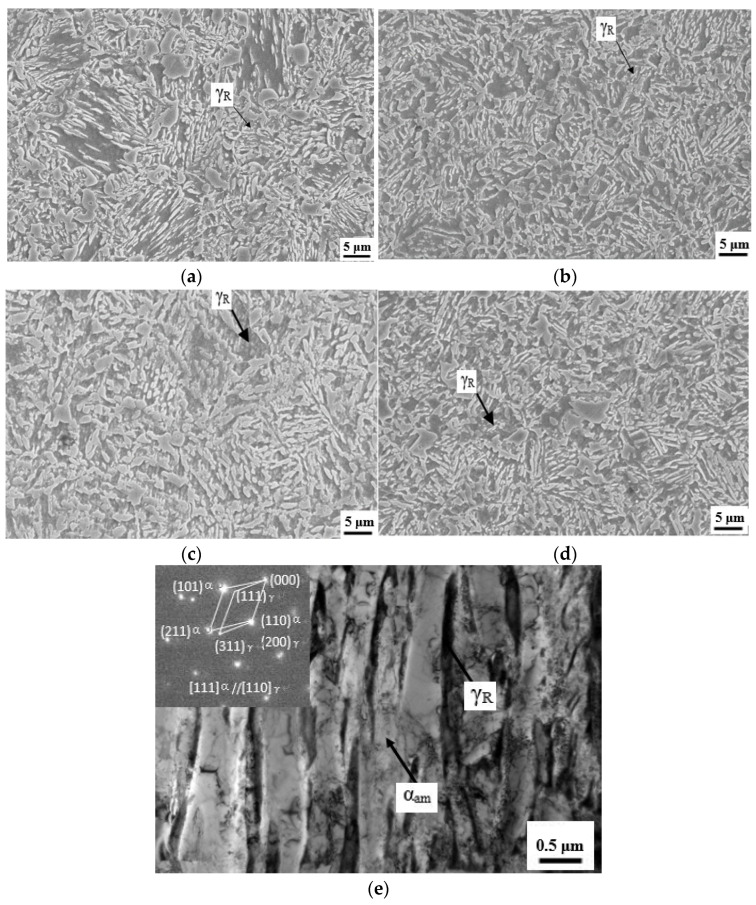
Microstructures of the studied steels observed by SEM showing annealed martensitic matrix and retained austenite. (**a**) TAM-V steel with 14.45% RA; (**b**) TAM-Ti steel with 12.96% RA; (**c**) TAM-Nb steel with 12.49% RA; (**d**) TAM-R steel with 14.42% RA; and (**e**) Transmission electron microscope (TEM) image of TAM-R steel showing retained austenite inserted into martensite laths.

**Figure 4 materials-14-07752-f004:**
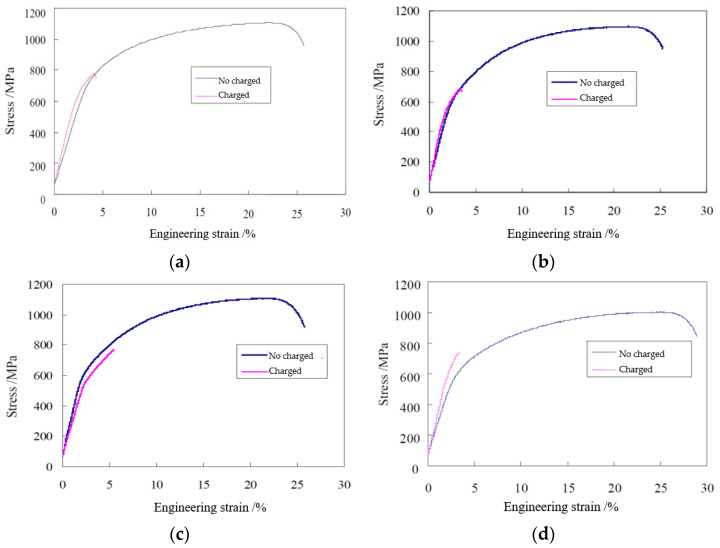
Engineering stress–strain curves of four studied steels during SSRT under conditions of with/without hydrogen charging. (**a**) TAM-V steel; (**b**) TAM-Nb steel; (**c**) TAM-Ti steel; and (**d**) TAM-R steel.

**Figure 5 materials-14-07752-f005:**
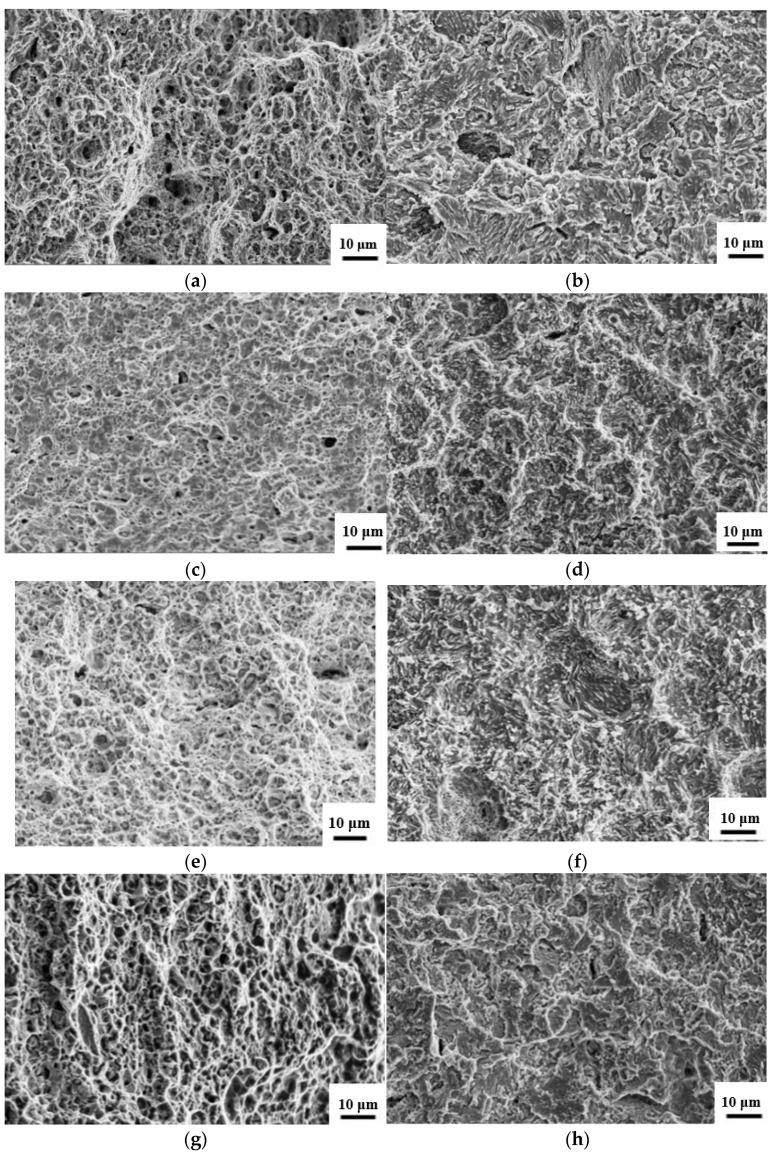
Fracture appearance of steels after SSRT test along with/without hydrogen charging. Clear dimples present in the left sides of the pictures showing a ductile fracture pattern, and cleavage faces seen in the right sides of the pictures implies the occurrence of the hydrogen induced embrittlement effect in steels. (**a**) TAM-V steel without charging; (**b**) TAM-V steel with charging; (**c**) TAM-Ti steel without charging; (**d**) TAM-Ti steel with charging; (**e**) TAM-Nb steel without charging; (**f**) TAM-Nb steel with charging; (**g**) TAM-R steel without charging; and (**h**) TAM-R steel with charging.

**Figure 6 materials-14-07752-f006:**
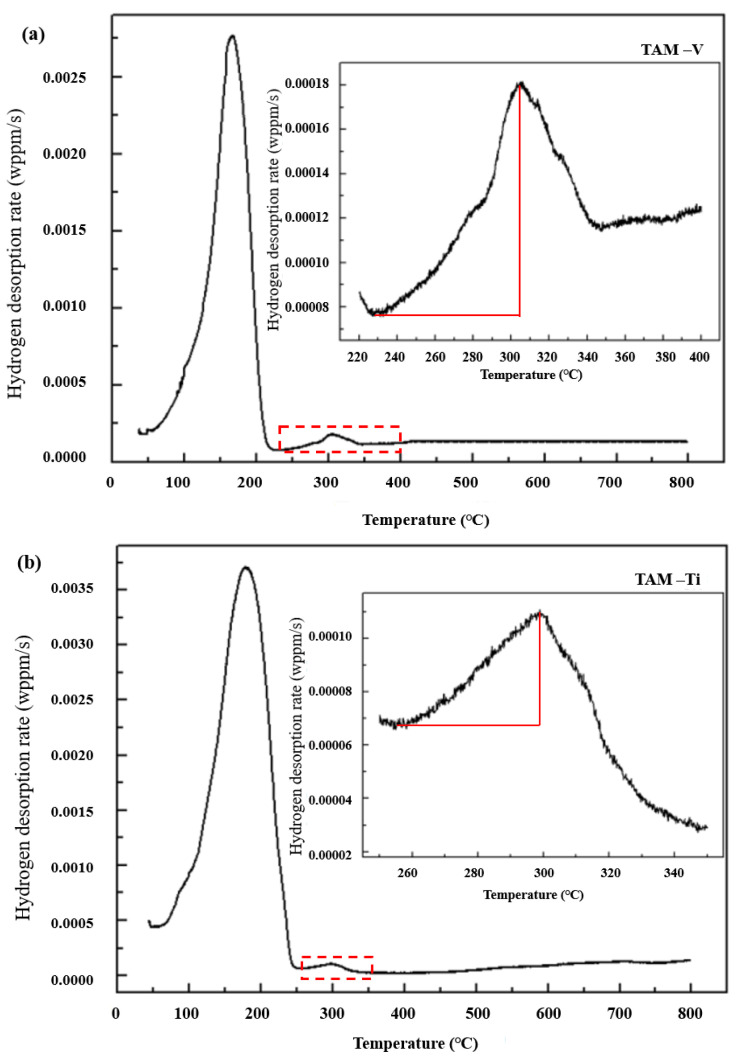
TDA curves of four kinds of hydrogen-charged steels during heating up to 800 °C. The incorporated charts are the enlarged second-peak section shown in the corresponding charts.

**Table 1 materials-14-07752-t001:** Chemical compositions of used steels in this study (wt.%).

Elements	C	Si	Mn	V	Ti	Nb
TAM-V	0.20	1.53	2.13	0.052	-	-
TAM-Ti	0.19	1.47	2.15	-	0.048	-
TAM-Nb	0.20	1.47	2.09	-	-	0.049
TAM-R	0.19	1.42	2.02	-	-	-

**Table 2 materials-14-07752-t002:** Slow strain rate tensile test results for four studied steels.

Steels	Applied Conditions	*R*_m_/MPa	*A*/%	*I_T_/%*	*I_D_/%*
TAM-V	No charged	1106	25.70	29.8	83.4
	Charged	776	4.26		
TAM-Nb	No charged	1095	24.90	37.4	87.3
	Charged	686	3.17		
TAM-Ti	No charged	1107	25.76	30.5	78.8
	Charged	769	5.45		
TAM-R	No charged	1002	28.93	24.2	91.1
	Charged	760	2.57		

**Table 3 materials-14-07752-t003:** Thermal desorption hydrogen content measured by TDS for hydrogen charging samples of four studied steels (ppm).

Samples	TAM-V	TAM-Ti	TAM-Nb	TAM-R
De-H during first peak section	8.66	9.43	9.34	10.02
De-H during second peak section	0.42	0.22	0.10	0.02

## Data Availability

The data presented in this study are not publicly available due to the agreement reached between the funding institution and project supervisor.

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
