# Peer review of "Mechanical Behaviors of Microalloyed TRIP-Assisted Annealed Martensitic Steels under Hydrogen Charging"

_materials, 2021, doi:10.3390/ma14247752_

Round 1

Reviewer 1 Report

The paper is very bad written. English language should be revised. There are a lot of grammatical mistakes. There are a lot of lack of spaces between words. All of them should be corrected.

Reviewer 2 Report

At this stage paper is informative but not of great interest to readers. It should be improved by better and more detailed explanation about microstructure and micoalloying elements effects on hydrogen role in reducing TS and elongation, i.e. ductility. Explanations offered in the text are too general  and well known. More micrographs and evidence are needed to explain detailed effects of different microalloying elements.

English should be carefully revised, not only what I marked in the attached file!

Round 2

Reviewer 1 Report

The authors must make clear in the test methodology the number of specimens used in each type of test.
To the question (made in the first review) of how many specimens they have used in each test condition, the authors reply that they have only used one specimen.
Do the authors believe that using only one specimen per condition they can assert that the results obtained are representative of the behavior of the tested materials?
In order to assert the stated conclusions, at least three tests must be carried out in each tested condition.

Author Response

Thanks for your comments. Please see the attachment regarding the modification and response to your question.

Reviewer 2 Report

Paper is now OK, corrections are sufficient, as well as added explanation.

Author Response

Thank you very much. Please see the revised manuscipt where English spelling and style are modified.